# Adapting Humanoid Locomotion over Challenging Terrain via Two-Phase Training

**Wenhao Cui** [*], **Shengtao Li** [*], **Huaxing Huang** [*], **Bangyu Qin, Tianchu Zhang, Jinchao Han,**
**Liang Zheng, Ziyang Tang, Chenxu Hu, Ning Yan, Jiahao Chen, Zheyuan Jiang** [*†]
Noetix Robotics, Beijing

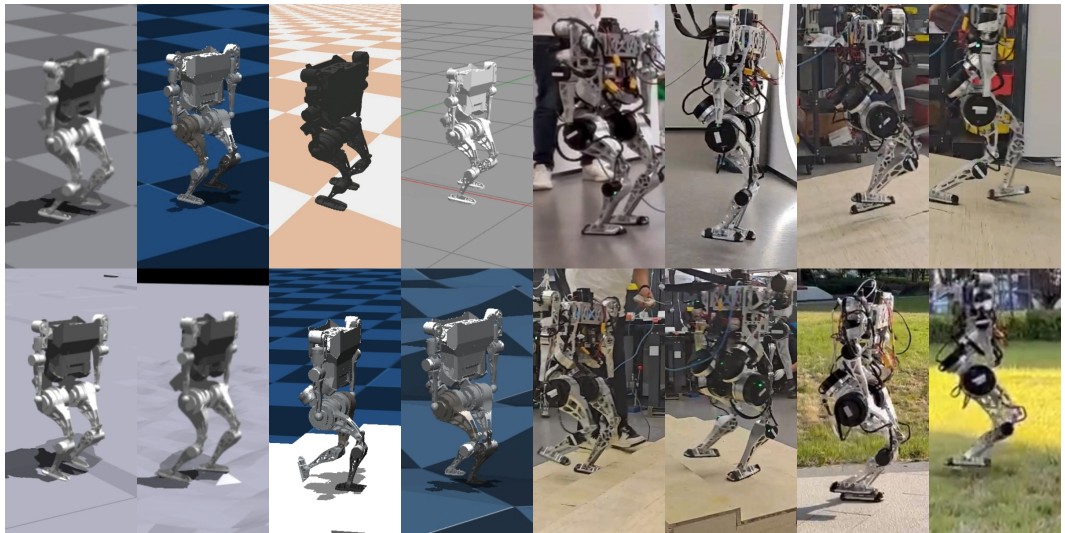

Figure 1: Our method enables the humanoid robot to perform stably across various simulated environments. We successfully transfer the policy to the physical robot in various real-world environments.

**Abstract:** Humanoid robots are a key focus in robotics, with their capacity to navigate tough terrains being essential for many uses. While strides have been made, creating adaptable locomotion for complex environments is still tough. Recent progress in learning-based systems offers hope for robust legged locomotion, but challenges persist, such as tracking accuracy at high speeds and on uneven ground, and joint oscillations in actual robots. This paper proposes a novel training framework to address these challenges by employing a two-phase training paradigm with reinforcement learning. The proposed framework is further enhanced through the integration of command curriculum learning, refining the precision and adaptability of our approach. Additionally, we adapt DreamWaQ to our humanoid locomotion system and improve it to mitigate joint oscillations. Finally, we achieve the sim-to-real transfer of our method. A series of empirical results demonstrate the superior performance of our proposed method compared to state-of-the-art methods.

**Keywords:** humanoid robots, locomotion, reinforcement learning, curriculum learning, sim-to-real

---

[*]Denotes equal contribution. Videos are available on https://sites.google.com/view/adapting-humanoid-locomotion/two-phase-training

[†]Correspondence to: `merlin.jiang@noetixrobotics.com`

8th Conference on Robot Learning (CoRL 2024), Munich, Germany.

# 1 Introduction

The humanoid robot has gained significant interest due to its human-like capabilities in performing various tasks [1, 2, 3]. Recent advances in reinforcement learning-based legged locomotion systems [4, 5, 6, 7] have shown great potential in dealing with complex, challenging environments. In this paper, we aim to address the challenges of achieving robust walking for humanoid robots on difficult terrains. We propose a two-phase sim-to-real (Sim2Real) learning framework, integrated with novel learning-based state estimation and curriculum learning, for humanoid locomotion.

The proposed framework divides the training process into two phases. In the first phase, the robots are trained on less challenging terrains with hand-crafted reference motion based on their structural and kinematic characteristics. Several reward functions are introduced to encourage the robots to imitate these motions quickly. A sinuous wave reference motion creates alternating leg movements for stable walking, encouraging quick imitation. However, these rewards may hinder exploration in phase two, which focuses on challenging terrains where adaptive gaits are necessary. Therefore, in phase two, by eliminating the reliance on predefined reference motions, our approach enables the robot to navigate more complex terrains with enhanced agility and efficiency.

Besides the two-phase training paradigm, we incorporate a curriculum learning strategy and a learning-based state estimation method into our framework. We adopt curriculum learning on velocity command tracking, as proposed by Margolis et al. [8]. This progressive expansion in velocity enables the robot to develop robust and wide-ranging command tracking capabilities, essential for the overall training process. Our experimental validation shows that our robot can achieve a maximum speed of 1 m/s on diverse terrains. Furthermore, inspired by the DreamWaQ method [9], we design an estimator network to estimate the robot's base velocity. Compared to the original method, we utilize multi-frame(20) observations rather than single-frame inputs to the actor policy.

We design qualitative and quantitative experiments with various baselines to evaluate the proposed approach, as shown in Fig. 1. Empirical results indicate that our method outperforms the baselines in terms of command tracking, task success rate, and robustness. The proposed approach demonstrates excellent robustness, even when transferred to novel terrains that are not initially present in the simulated environments. Our contributions can be summarized as follows:

- We propose a simple yet effective two-phase training paradigm that enables us to learn robust, relatively high-speed locomotion controllers to traverse various challenging terrains on a real humanoid robot.
- We improve the DreamWaQ method by changing the input for the policy network from single-frame to multi-frame(20) information. This straightforward yet practical approach enhances walking stability and significantly benefits the Sim2Real transfer.
- We adopt a velocity command curriculum to our training framework, enabling the robot to work at a wide range of velocity commands over challenging terrains.

# 2 Related Work

**Reinforcement Learning for Legged Locomotion**. Deep reinforcement learning has shown great potential in legged locomotion control and is becoming the mainstream control algorithm for legged robots. Applying the teacher-student training paradigm in Sim2Real learning of legged locomotion is popular among research works. Lee et al. [10] applied teacher-student training to the quadruped robot ANYmal, resulting in a robust controller capable of traversing challenging terrains. Kumar et al. [11] trained locomotion policies with rapid motor adaptation, enabling them to adapt to environmental changes quickly. The teacher-student training paradigm applies a neural network to perform system identification. In contrast, another stream of research uses a separate neural network to perform state estimation, which is relevant to this work. Ji et al. [12] trained a state estimator policy to estimate robot base velocity and feet contact. DreamWaQ [9] further extends learning based state estimation into estimating hidden states. Learning with reference motion can promote the training

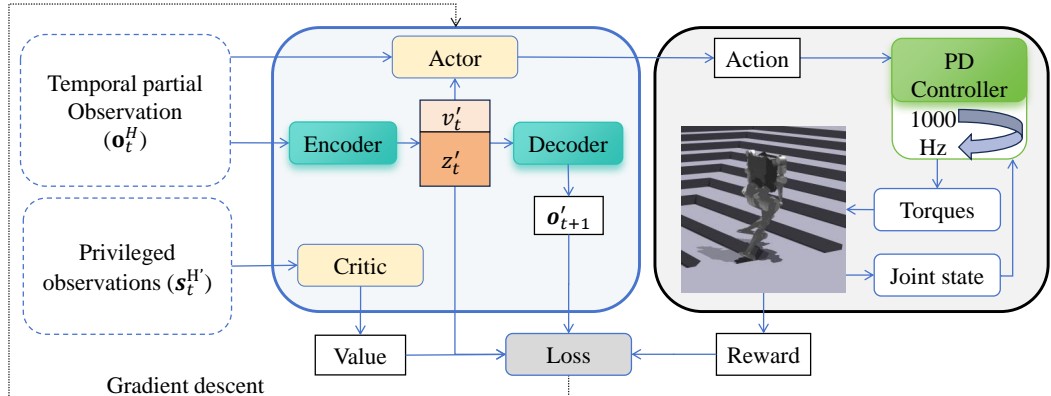

Figure 2: **Overview of our method**: The temporal partial observation $\mathbf{o}_t^H$ is encoded by an encoder which outputs the velocity estimation $\mathbf{v}_t'$ and the latent $\mathbf{z}_t'$ to be received by actor-network and decoder. The decoder predicts the observation at the next time step, denoted by $\mathbf{o}_{t+1}'$, combining with $\mathbf{o}_{t+1}$ obtained from the simulation to calculate the loss. The details of loss function designing are in Sec 3.3. Action from the actor network presents the joints' target position; then the PD Controller calculates the torques. This framework applies to both phases.

process and make the resulting gait more natural. Peng et al. [13] introduced the use of Deep Mimic [14] to learn robotic locomotion skills by imitating animals. Escontrela et al. [15] further utilized Adversarial Motion Priors (AMP) to train control policies for a quadrupedal robot, highlighting that AMP can effectively substitute complex reward functions. Yang et al. Bohez et al. [16] trained a low-level locomotion controller for a quadruped robot by imitating actual animal data for various tasks. Curriculum learning, which can help with exploration, is essential in legged robotics. Margolis et al. [17] applied curriculum learning to train a locomotion controller for the Mini Cheetah robot, enabling it to achieve speeds of up to 3.9m/s, surpassing traditional controllers' speeds by a large margin. Other notable works include directly learning locomotion skills in the real world [18, 19], learning from both proprioceptive observation and exteroceptive observation [20, 21, 22], applying unsupervised skill discovery to discover locomotion skills [23].

**Reinforcement Learning for Humanoid Locomotion**. Humanoid locomotion is one of the most challenging tasks in legged locomotion and has been getting increased attention in recent years. Due to the embodiment difference and the inherent instability of bipedal robots, algorithms that work well on quadruped robots may need improvements to function fully on a bipedal robot. Kumar et al. [7] extended the A-RMA algorithm to the bipedal robot Cassie. Learning-based state estimation is further enhanced and studied in [24]. Imitating human motion is also of significant attention in bipedal locomotion. Cheng et al. [25] relaxes imitation constraints on legs and focuses on imitating the upper body movements of a humanoid robot. Zhang et al. [26] introduced AMP into humanoid locomotion. Other works include designing appropriate reward functions [27], using transformers to represent locomotion policy [28], Sim2Real learning of humanoid control policy [29].

## 3 Method

The overall architecture of our method can be depicted as illustrated in Fig. 2. It provides a schematic representation of the network's composition, detailing the interconnected layers and components that constitute the system.

### 3.1 Network Input

**Estimator Network**: Similar to the works from Kumar et al. [11] and Margolis et al. [8], an estimator network using the history observation $\mathbf{o}_t^H = [\mathbf{o}_t \ \mathbf{o}_{t-1} \ \dots \ \mathbf{o}_{t-H}]^T$, is necessary for estimating the latent variable. In contrast to Nahrendra et al. [9], we utilize $\mathbf{o}_t^H$ as part of the input for the policy network, rather than inputting a single frame of observation $\mathbf{o}_t$, demonstrating considerably superior performance. The single observation $\mathbf{o}_t$ is a 40-dimensional vector encom-

passing a rich set of sensory inputs and robot dynamics. It is an $n \times 1$ vector defined as follows: $\mathbf{o}_t = [\mathbf{p}_t \quad \mathbf{c}_t \quad \mathbf{y}_t \quad \mathbf{q_t} \quad \dot{\mathbf{q}}_t \quad \mathbf{a}_{t-1}]^T$ , Where $\mathbf{p}_t$ (2D) is the current phase indicated a regular motion which is generated by a sinusoidal signal, $\mathbf{c}_t$ (3D) is a velocity commands used to give the current desired speed direction of the body, $\mathbf{y}_t$ (5D) is the body angle (only roll and pitch used) and angular velocity, $\mathbf{q_t}$ (10D) is the amount by which the joint angles changes relative to the default joint angles, $\dot{\mathbf{q}}_t$ (10D) is the joint angular velocity, and $\mathbf{a}_{t-1}$ (10D) is the previous action.

**Policy Network**: The latent variable obtained from the output of the estimator network are $\mathbf{z}_t$ and body velocity $\mathbf{v}_t$, and then, they are coated together with the temporal proprioceptive observation $\mathbf{o}_t^H$ as the input of the policy network. Finally, the complete policy network, parameterized by $\phi$, can be expressed as $\pi_\phi(\mathbf{a}_t | \mathbf{o}_t^H, \mathbf{z}_t, \mathbf{v}_t)$.

**Value Network**: The value network receives more information from the world's state, which can be obtained directly from the simulation environment. The privileged observation $\mathbf{s}_t$ is defined as: $\mathbf{s}_t = [\mathbf{o}_t \quad \mathbf{v}_t \quad \mathbf{h}_t \quad \mathbf{f}_t]^T$ , where $\mathbf{h}_t$ and $\mathbf{f}_t$ means the topographical scans. Specifically, $\mathbf{h}_t$ denotes the terrain map of the robot's surroundings, while $\mathbf{f}_t$ represents a localized terrain map for the robot's left and right feet.

## 3.2 Reward

The design of the reward function is pivotal in shaping the agent's policy, as it directly influences the actions selected and the resulting motion patterns exhibited by the humanoid robot. In humanoid locomotion, the reward signal reflects the efficiency and effectiveness of movement and embodies the safety and adaptability required for navigating unpredictable terrains and situations.

In this work, we develop a reward system that propels the humanoid towards achieving agile, stable, and energy-efficient locomotion across diverse environments and tasks. The rewards delineated in [9, 29] facilitate the acquisition of basic ambulation; however, the stability of the robot's movements leaves room for improvement. The reward architecture comprises several components:

$$ r = r_{gait} + r_{command} + r_{root} - r_{energy} + p \cdot r_{reference} $$

where $r_{gait}$ regulates gait patterns, $r_{command}$ promotes speed commands tracking, $r_{root}$ ensures the upright posture and correct orientation of the robot's base link, $r_{energy}$ penalize excessive energy consumption, $r_{reference}$ encourages adherence to reference motion, $p \in \{0, 1\}$ equals one only in training phase one. The detailed reward function and weights are in Table 4.

## 3.3 Loss Function

For estimator, similar to Nahrendra et al. [9], in our approach, we require an encoder-decoder framework where the decoder generates $\mathbf{o}'_{t+1}$ to compute the encoder's loss explicitly. The specific method involves calculating the mean-squared-error (MSE) loss between the estimated body velocity $\mathbf{v}'_t$ and the ground truth $\mathbf{v}_t$ from the simulator. Additionally, we employ the standard $\beta$-VAE loss to compute the loss for decoder result $\mathbf{o}'_{t+1}$ and the ground truth $\mathbf{o}_{t+1}$, and sum these two losses. To further increase the sim-to-real robustness of the learned policy, we also utilize adaptive bootstrapping (AdaBoot), a technique similar to that used in Nahrendra et al. [9], to stabilize the estimator's updates.

$$ L_{\text{est}} = L_{\text{MSE}} + L_{\text{VAE}} $$
$$ L_{\text{MSE}} = \text{MSE}(\mathbf{v}_t, \mathbf{v}'_t), \quad L_{\text{VAE}} = \text{MSE}(\mathbf{o}_{t+1}, \mathbf{o}'_{t+1}) + \beta \text{KL}(g(\mathbf{z}_t | \mathbf{o}_{t+1}^H) \| p(\mathbf{z}_t)) $$

## 3.4 Learning System

**Two-phase training**: The training is divided into two phases. In the first phase, the reference motion is incorporated as a reward to encourage the robot to learn to move quickly according to a standard sinusoidal gait. In the second phase, the reward for the reference motion is removed to allow the robot to learn better strategies to adapt to the set terrain. The terrain includes planes, steps, slopes, and stairs. During the first phase, the difficulty of the terrain is relatively low, with more simple

terrains and lower stairs height. However, in the second phase, we increase the difficulty of the terrain, raising the proportion of high-difficulty terrains such as steps and stairs to 50% with higher stairs height. In this way, our robot can step up to 16cm high stairs at most.

**Commands Curriculum Learning**: During training, we observed that robots tend to walk in place on dangerously uneven terrains, even with high linear velocity tracking rewards. To address this, inspired by Margolis et al. [8], we implemented a curriculum learning strategy for velocity, which means the robot initially focuses on learning to follow the reference gait phase at low speeds. Once the average velocity reward exceeds a threshold $r = 0.75$, we gradually increase the linear velocity commands by $\triangle v_{x,y} = 0.05$ m/s and angular velocity commands by $\triangle \omega_z = 0.05$ rad/s, enabling effective speed tracking.

**Multi-Observation with Estimation**: We have identified that single-frame observation augmented with estimator-predicted information [9] [24], while capable of tracking velocity commands in simulation, often result in joint tremors and suboptimal performance on varied terrains. Conversely, relying on multi-frame(20) observation without estimator [29] integration leads to slow movement and weak action robustness. To address these issues, we developed our proprietary "Multi-Observation with Estimation" method to train policies. This approach effectively tracks velocity commands, navigates through diverse terrains, and demonstrates excellent robustness and adaptability when deployed on actual hardware. It seamlessly transfers the policy from simulation to the real robot.

### 3.5 Sim-to-Real

To accomplish the sim-to-real transfer, we designed various randomizations and delays for instructions. Firstly, we applied randomizations to the observation and the robot's properties for domain randomization. In order to mimic the effects of real-world deployment as closely as possible, we introduced a random delay of up to 15% in both action and torque to help bridge the sim2real gap. Besides, We adjusted the center of gravity offset and link mass to adapt to the gap in the real environment. Additionally, we apply a random force to each link every time an action is outputted. These measures have significantly improved the situation where the robot's hardware undergoes some design adjustments, which necessitated retraining with past methods. A detailed summary of the domain randomization and the execution delays is presented in Appendix Table 5.

## 4 Experimental Results

We conduct Numerous comparative and ablation studies in simulation and reality. This section will provide a detailed account of the specific experimental designs and the final comparative results.

The comparative experiments we performed are as follows: *Baseline*: a no estimator policy with frame stacks observation; *DreamWaQ*: the reproduced method presented in Nahrendra et al. [9]; *A-RMA*: the reproduced method presented in Kumar et al. [7]; *Ours w/o CC*: Our policy without commands curriculum, fixed at [-1, 1] m/s; *Ours w/o Ref*: Our policy without rewards associated to reference motion; *Ours w/o LVAE*: Our policy without LVAE loss; *Phase 1*: Our training strategy for the first phase with reference motion.

**Reference motion**: In experiment *Ours w/o Ref*, the significant differences in the contact forces between the robot's two feet can cause considerable strain on its motors and links, resulting in significant wear and tear, as shown in Fig. 3, the figure illustrates the contact forces for both the left and right feet, highlighting a stark contrast between the policy executed with reference motion and one without it. The latter demonstrates a substantial variability in force application, an issue absent in our policy. The absence of reference motion can result in imbalances, potentially leading to motor damage in real robots, whereas our approach ensures consistent stability and motor protection. Additionally, the uncoordinated movements significantly increase the likelihood of the robot falling, underscoring an urgent need for synchronized actions and balanced force distribution to enhance reliability and prevent such incidents. Consequently, we omitted this comparison in the following analysis.

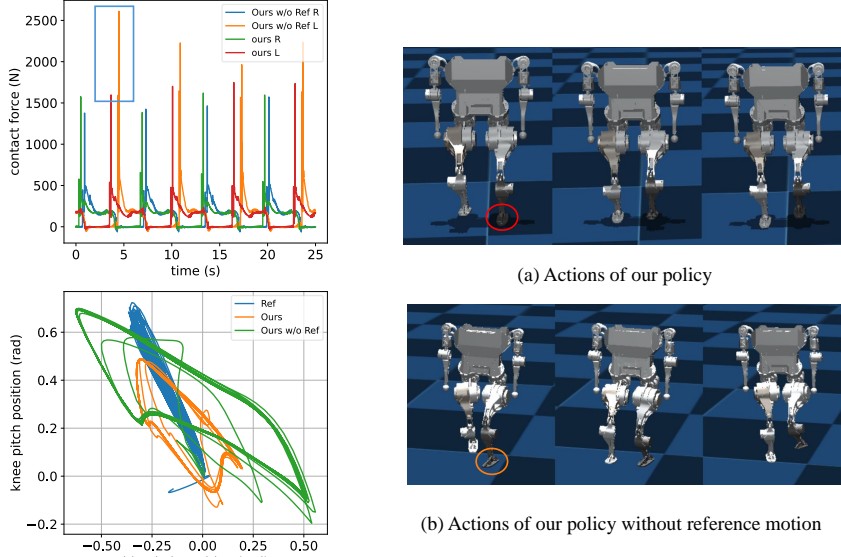

(a) Actions of our policy

(b) Actions of our policy without reference motion

Figure 3: The upper-left diagram reveals that the ground contact forces produced by the strategy, which lack reference motions, surpass the endurance limits of the motors extremely. The lower-left illustration indicates that the hip and knee joints, under the no-reference strategy, are inclined to move to more extreme positions, potentially causing damage to real robots. Right-side comparison strikingly outlines the difference in gaits between those informed by reference motions and those without, underscoring the no-reference strategy's challenge in sustaining stability.

**Multi-task**: We conducted experiments comparing the robot's performance between Phase one and two. In the first experiment, the robot was tested for its ability to climb stairs at a speed of 1 m/s, with success defined as clearing a six-step staircase. The second experiment evaluated the robot's ability to traverse slopes with varying gradients at the same speed. Our approach, not constrained by predefined reference motion, showed effective traversal of steps and slopes, as detailed in Table 1. In contrast, other strategies exhibited instability on terrains deviating from reference motion parameters. While alternative methods struggled to climb steps over 5 cm high, our system maintained an 85% success rate on 10 cm steps. Similar proficiency was observed in traversing slopes steeper than 15 degrees, with Phase two being the only one to navigate them effortlessly.

Table 1: The rate of successful stair ascents, evaluated across a spectrum of increasing stair heights, correlates with the performance on inclines of varying degrees, showcasing the adaptability of the system to diverse elevation challenges.

| | Sim | | | | Real | | | |
|---|---|---|---|---|---|---|---|---|
| | Phase 2 | Phase 1 | D'WaQ | A-RMA | Phase 2 | Phase 1 | D'WaQ | A-RMA |
| 0.03m | 1.00 | 0.75 | 0.70 | 0.65 | 1.00 | 0.70 | 0.75 | 0.55 |
| 0.05m | 1.00 | 0.15 | 0.15 | 0.20 | 0.95 | 0.05 | 0.10 | 0.10 |
| 0.08m | 0.95 | 0.05 | 0 | 0.05 | 0.85 | 0 | 0 | 0 |
| 0.10m | 0.85 | 0 | 0 | 0 | 0.80 | 0 | 0 | 0 |
| 5° | 1.00 | 0.95 | 0.85 | 0.70 | 1.00 | 0.95 | 0.75 | 0.70 |
| 10° | 1.00 | 0.80 | 0.85 | 0.55 | 1.00 | 0.85 | 0.8 | 0.40 |
| 15° | 1.00 | 0.60 | 0.45 | 0.30 | 0.95 | 0.45 | 0.2 | 0.25 |
| 20° | 0.95 | 0 | 0 | 0 | 0.95 | 0 | 0 | 0 |

**Estimator loss**: The estimator loss plays a crucial role in assessing the robot's ability to accurately predict linear velocity and latent vectors, which significantly aids in our ability to follow speeds and traverse various terrains effectively. We recorded all the loss curves of the methods mentioned

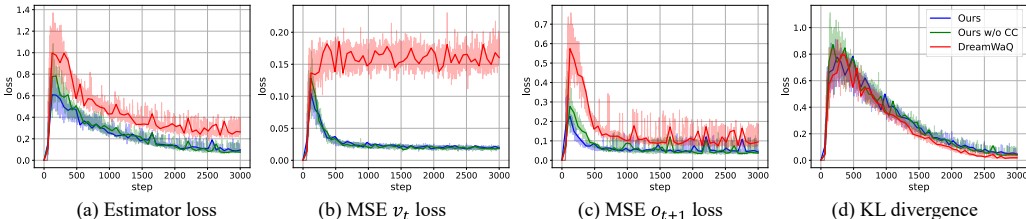

(a) Estimator loss  (b) MSE $v_t$ loss  (c) MSE $o_{t+1}$ loss  (d) KL divergence

Figure 4: Without our policy, the predictions for $\mathbf{v}_t$ rendered by the DreamWaQ exhibit a significantly inferior performance when juxtaposed with our approach. Furthermore, the subsequent $\mathbf{o}_{t+1}$ generated by DreamWaQ also needs to catch up in comparison to the outcomes yielded by our strategic implementation. The KL divergence across policies indicates a comparable level of learning. However, upon a comprehensive assessment, our strategy demonstrates superior performance in estimating the velocity $\mathbf{v}_t$ and latent variables $\mathbf{z}_t$.

above. The result is shown in Fig. 4. It shows that our Multi-Observation with Estimation policy demonstrates superior predictive capabilities for $\mathbf{v}_t$ and $z_t$ compared to DreamWaQ, which relies solely on single observation within its actor policy.

**Stability**: We conducted a comprehensive empirical analysis comparing our proposed strategy alongside DreamWaQ and a standard baseline across a continuum of simulated and actual robotic platforms. This battery of tests was designed to rigorously evaluate the adaptability and versatility of each algorithm in diverse environmental conditions. Furthermore, we subjected the algorithms to a series of perturbations that simulate the unpredictable nature of real-world conditions, thereby evaluating their resilience and robustness. This final testing phase is instrumental in gauging the practical applicability and reliability of the algorithms under non-ideal circumstances.

While all robotic systems encountered increased challenges on low-$\mu$ surfaces, precipitating slippage and an elevated risk of collapse, our strategy adeptly traversed the array of terrains as shown in Table 2 In the context of disturbance rejection trials, the intrinsic instability of the alternative algorithms led to an almost inevitable collapse under lateral forces and suboptimal success rates from anterior and posterior forces. Conversely, our strategy consistently exhibited unparalleled stability, withstanding forces applied across all directional vectors, resulting in Table 3. Appendix B shows that our robotic system has demonstrated the ability to traverse various terrains and obstacles.

Table 2: Success rate of passing various terrain.

|  | **Ours** | **A-RMA** | **D'WaQ** | **Ours w/o LVAE** | **Baseline** |
|---|---|---|---|---|---|
| Urban | 1.00 | 0.70 | 0.75 | 0.90 | 0.70 |
| PVC floor | 0.90 | 0.20 | 0.10 | 0.00 | 0.00 |
| Grasslands | 1.00 | 0.40 | 0.35 | 0.15 | 0.55 |
| Wired-strewn | 0.85 | 0.00 | 0.00 | 0.00 | 0.00 |

Table 3: Success rate of stability under external force.

|  | **Ours** | **A-RMA** | **D'WaQ** | **Ours w/o LVAE** | **Baseline** |
|---|---|---|---|---|---|
| Front | 1.00 | 0.45 | 0.40 | 0.40 | 0.10 |
| Side | 0.85 | 0.05 | 0.20 | 0.15 | 0.00 |
| Back | 0.90 | 0.20 | 0.05 | 0.35 | 0.05 |

**Commands Tracking**: The robotic system's primary objective is to track the velocity commands issued to it accurately. Consequently, the efficacy of commands tracking emerges as a pivotal metric for evaluating the success of our experimental endeavors. This performance is vividly illustrated in Fig. 5. Observation have elucidated that when faced with arduous terrains, including stairs with heights exceeding 6 cm or a 15-degree inclined uneven terrain within our training set. Baseline, DreamWaQ and A-RMA tend to prioritize safety, consequently adopting a more cautious and slowed pace. Furthermore, the variant of our approach that forgoes a commands curriculum exhibits a di-

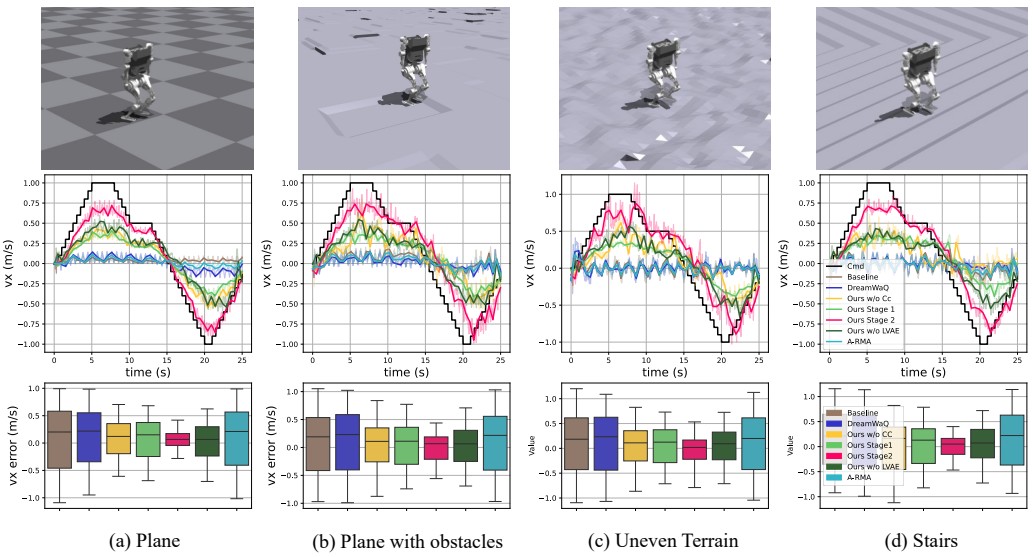

Figure 5: The first tier of images presents a compendium of scenarios for testing, encompassing a flat plane, a plane with obstacles, uneven terrain, and a staircase. The second stratum delineates velocity profiles, illustrating the strategy's actual performance in tracking predefined commands trajectories, represented by a black line. The third layer encompasses box plots that depict the distribution of velocity tracking errors within the simulated environment, the horizontal line indicating the median value.

minished capacity for velocity tracking, potentially due to the vast exploration space overwhelming the actor's learning process. Additionally, ours without LVAE have indicated that the Euclidean distance of the observations may not accurately represent the discrepancy from the target, resulting in suboptimal walking posture and inadequate speed following. In striking contrast, our refined approach showcases an enhanced aptitude for swiftly and adeptly acquiring and adjusting to velocity directives. Consequently, our strategy must incorporate a commands curriculum to ensure proficient learning and adaptation to the velocity commands.

## 5   Limitation

Our policy, though effective in its intended tasks, is limited by the lack of perception for terrain movements. To address this, we plan to integrate perception systems into our future work, thereby enhancing our strategies' adaptability to more complex challenges and expanding the operational capabilities of our robotic systems. Additionally, we recognize our approach has limitations, particularly in parameter tuning for diverse robots, which is a complex engineering task. Adjusting reward parameters based on a robot's size and carefully selecting domain randomization parameters are essential; for instance, center of mass randomization exceeding 0.06 cubic meters can severely impact real-world performance. Time-related parameters need finely tuned to match the anatomical proportions of different robots to ensure optimal dynamic responses. To address these challenges and improve our strategies' adaptability, we plan to investigate solutions such as imitation learning, aiming to enhance our system's robustness and applicability across various robotic platforms.

## 6   Conclusion

In this paper, we have presented a comprehensive study on the enhancement of steady humanoid robot locomotion, achieved through the refinement of estimator and actor policies. This research has broad implications, establishing a foundation for versatile and adaptive locomotion across varied terrains and obstacles through a two-phase training method that expands the robot's gait capabilities. Additionally, the study demonstrates an exceptional velocity tracking capability with a velocity commands curriculum that underscores the system's precision and responsiveness.

**Acknowledgments**

We extend our heartfelt gratitude to the anonymous reviewers for their insightful feedback, which significantly contributed to the improvement of this manuscript. This work was supported by the Noetix Robotics.

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

# A DETAILED EXPERIMENT SETUP

## A.1 REWARD

In this section, we detail the reward function and the weights. The reward function is comprised of several critical components, each serving a distinct purpose. The exponential function is represented by $\exp(\cdot)$, and the variance function is denoted by $\mathrm{var}(\cdot)$. The terms $(\cdot)^{\mathrm{des}}$ and $(\cdot)^{\mathrm{cmd}}$ are used to denote the desired and commanded values, respectively. The robot's body frame is defined by the coordinates $x$, $y$, and $z$, with $x$ and $z$ oriented in the forward and upward directions. The rotation angles of the robot's coordinate system are given by roll, yaw, and pitch. $P_{f(t)}, I_{d(t)}, I_{c(t)}, T_{\mathrm{air}}, v, \omega, h, d_f, d_k, g, \theta, \tau$ represent the height of the foot at time t, the phase at time t in the gait cycle, the contact status of the foot at time t, the aerial time of the foot, the linear velocity of the robot's root, yaw rate, height, foot distance, knee distance, the projection of the gravity vector onto the robot's body frame, joint position, and joint torque.

Table 4: Setup of reward function and scales.

| Reward | Equation ($r_i$) | Scale ($w_i$) |
|---|---|---|
| Feet clearance | $\left(p_{f(t)}^{\mathrm{des}} - p_{f(t)}\right)^2 \cdot \left(1 - I_{d(t)}\right)$ | -0.01 |
| Feet air time | $T_{air} \cdot I_{c(t)}$ | -0.001 |
| Follow gait phase | $\left(1 - I_{d(t)}\right) \cdot I_{c(t)}$ | -0.001 |
| Feet slip | $\omega \cdot I_{c(t)}$ | -0.005 |
| Feet&Knee distance | $\frac{\exp\{-100\cdot|0.3-d_{f,k}|\}+\exp\{-100\cdot|0.125-d_{f,k}|\}}{2}$ | 0.4 |
| Lin. velocity tracking | $\exp\left\{-4\left(\mathbf{v}_{xy}^{\mathrm{cmd}} - \mathbf{v}_{xy}\right)^2\right\}$ | 2.4 |
| Ang. velocity tracking | $\exp\left\{-4\left(\omega_{\mathrm{yaw}}^{\mathrm{cmd}} - \omega_{\mathrm{yaw}}\right)^2\right\}$ | 1.1 |
| Velocity mismatch | $\frac{\exp\{-10(-\mathbf{v}_z)^2\}+\exp\{-5(-\omega_{\mathrm{roll, pitch}})^2\}}{2}$ | 0.5 |
| Orientation | $\|\mathbf{g}\|^2$ | 1.0 |
| Feet orientation | $\|\mathbf{g}_{\mathrm{feet}}\|^2$ | 1.0 |
| Default joint | $\exp\left\{-2\left(\theta - \theta_{\mathrm{zero}}\right)^2\right\}$ | 0.5 |
| Body height | $\left(h^{\mathrm{des}} - 0.6505\right)^2$ | -1.0 |
| Root accelerations | $\exp\left\{-\left(\ddot{\boldsymbol{\theta}}_{\mathrm{root}}\right)^3\right\}$ | 0.2 |
| Joint accelerations | $\ddot{\boldsymbol{\theta}}^2$ | $-1 \times 10^{-6}$ |
| Joint velocity | $\dot{\boldsymbol{\theta}}^2$ | $-5 \times 10^{-3}$ |
| Joint power | $\boldsymbol{\tau}^2$ | $-1 \times 10^{-5}$ |
| Action rate | $\left(\mathbf{a}_t - \mathbf{a}_{t-1}\right)^2$ | $-0.01$ |
| Smoothness | $\left(\mathbf{a}_t - 2\mathbf{a}_{t-1} + \mathbf{a}_{t-2}\right)^2$ | $-0.01$ |
| Joint position tracking | $\exp\left\{-2\left(\theta - \theta_{\mathrm{target}}\right)^2\right\}$ | 3.2 |

## A.2 DOMAIN RANDOMIZATION

We leverage domain randomization during training to narrow the reality gap. Specifically, we set the range of parameters as shown in Table 5, mainly consisting of delay of action and torque, randomization of position, velocity, friction, KP/KD factor, and CoM.

## A.3 IMPLEMENTATION DETAILS

Our humanoid robot, named N1, is equipped with a total of 18 degrees of freedom. In this work, we have immobilized the 8 joints associated with the arms, focusing exclusively on the 10 joints related

Table 5: Overview of Domain Randomization. Presented are the domain randomization terms and the associated parameter ranges. Additive randomization increments the parameter by a value within the specified range while scaling randomization adjusts it by a multiplicative factor from the same range.

| Parameter | Unit | Range | Operator | Type |
|---|---|---|---|---|
| Joint Position | rad | [-0.05, 0.05] | additive | Gaussian (lo) |
| Joint Velocity | rad/s | [-1.5, 1.5] | additive | Gaussian (lo) |
| Angular Velocity | rad/s | [-0.2, 0.2] | additive | Gaussian (lo) |
| Linear Velocity | m/s | [-0.1, 0.1] | additive | Gaussian (lo) |
| Euler Angle | rad | [-0.06, 0.06] | additive | Gaussian (lo) |
| Action Delay | ms | [0, 10] | - | Uniform |
| Torque Delay | ms | [0, 10] | - | Uniform |
| Friction | - | [0.1, 2.0] | - | Uniform |
| Kp factor | % | [80, 120] | scaling | Gaussian (lo) |
| Kd factor | % | [80, 120] | scaling | Gaussian (lo) |
| Motor Strength | % | [80, 120] | scaling | Gaussian (lo) |
| Payload | kg | [-5, 5] | additive | Gaussian (lo) |
| CoM | m | [-0.02, 0.02] | additive | Gaussian (lo) |
| Link Mass | % | [0.9, 1.1] | scaling | Gaussian (lo) |

to the legs. The motors' hip (pitch) and knee joint torque can reach up to 150 Nm, while the motors' torque of the foot joints is 36 Nm. This robot's total height and weight are 0.95 m and 23 kg.

Our RL control strategy operates at 100 Hz, coupled with an internal PD controller that runs at 1000 Hz. It ensures synchronization with the operational frequency of the actual hardware. We employ Isaac Gym for training and conduct sim-to-sim validation in various simulation environments, including MuJoCo, PyBullet, and Gazebo. This multi-environment approach ensures the robustness and adaptability of our models. We utilize the Proximal Policy Optimization (PPO) algorithm [30]. The details of our training parameters are presented in Table 6, where we outline the specifics that contribute to the enhanced performance of our model.

Table 6: Training hyperparameters

| Parameter | Value |
|---|---|
| Number of environments | 4096 |
| Training epochs | 2 |
| Learning rate | $10^{-5}$ |
| Gamma $\gamma$ | 0.995 |
| Lambda $\lambda$ | 0.95 |
| Batch size | 24 |
| Episode Length | 3000 |
| Backbone hidden layers | [512, 512, 128] |
| Encoder hidden layers | [768, 256, 64] |
| Activation function | ELU |
| Decoder observation | 19 |
| Number of Observation frame stack | 5 |
| Number of Privileged frame stack | 3 |
| Number of Aggregated Observation | 219 |
| Number of Aggregated Privileged Observation | 846 |

# B ALGORITHM

We present our complete algorithmic process in the following algorithm, where four networks are continuously updated through sampling from the simulation based on the process outlined below.

---

**Input:** Encoder network $est_\phi$, decoder network $dec_\varphi$, policy network $\pi_\theta$, value function $V_\psi$, environment, number of epochs $E$, mini-batch size $M$

**Output:** Optimized encoder network $est_\phi$, policy network $\pi_\theta$
Initialize encoder network $est_\phi$, decoder network $dec_\varphi$, policy network $\pi_\theta$ and value function $V_\psi$
Initialize advantages $A_t = 0$ and value targets $V_t = 0$
**for** *each epoch $e = 1$ to $E$* **do**
 **for** *each mini-batch $b = 1$ to $M$* **do**
  Initialize observation $o_0$ , obtain frame stack observation $o_0^H$ and collect rollouts
  **for** *each step $t = 1$ to $T$* **do**
   Compute linear velocity $v_t$ and latent features $z_t \sim est_\phi(\cdot|o_t^H)$
   Compute action $a_t \sim \pi_\theta(\cdot|o_t^H, v_t, z_t)$
   Execute $a_t$ in the environment and observe $o_{t+1}, o_{t+1}^H, r_t, d_t$
   Compute advantage estimate $\hat{A}_t$, value target $\hat{V}_t$ linear velocity target $\hat{v}_t$ and
    next step observation target $\hat{o}_{t+1}$; Update advantages $A_t = \rho_t A_{t-1} + \hat{A}_t$ and
    value targets $V_t = \gamma V_{t-1} + \hat{V}_t$
  **end**
  Perform multiple PPO updates using $A_t$ and $V_t$ to optimize $\pi_\theta$ and $V_\psi$ and estimator
   updates using $o_{t+1}$, $z_t$ and $v_t$ to optimize $est_\phi$ and $dec_\varphi$;
 **end**
**end**
**return** the optimized policy network $\pi_\theta$ and the encoder policy network $est_\phi$;

---

**Algorithm 1:** Training algorithm

# C GENERALIZATION

This section presents a series of experimental images that vividly illustrate our algorithm's intricate implementation details and robust generalization capabilities across various initial poses for robotic walking tasks, shown in the Fig. 6. We have achieved smooth locomotion across a spectrum of initial postures, as evidenced by the images depicting the robot's initial stance.

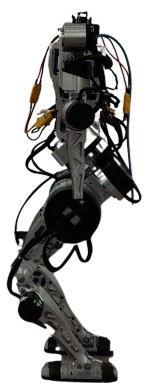 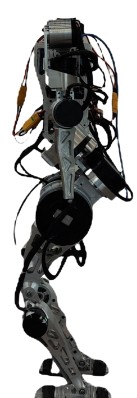 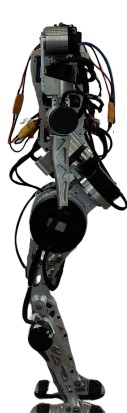

Figure 6: Training with diverse initial poses: An illustrative analysis of robotic locomotion across varied starting configurations.

Within the simulated environment, the Fig. 7 display the robot's upright and stable gait, underscoring the algorithm's exceptional precision in control within the virtual platform. We have further substantiated the algorithm's reliability, shown in Fig. 8, and practical utility by advancing to real-world scenarios. The deployment of our algorithm on an actual robot has endowed it with the ability to navigate in various straight-legged postures, as illustrated by the images portraying the robot's commendable equilibrium and stability. This rigorous process is a testament to the algorithm's resilience and adaptability.

These outcomes furnish compelling validation for our algorithm's ongoing refinement and application, simultaneously presenting innovative perspectives and methodologies that hold significant promise for future research within related domains.

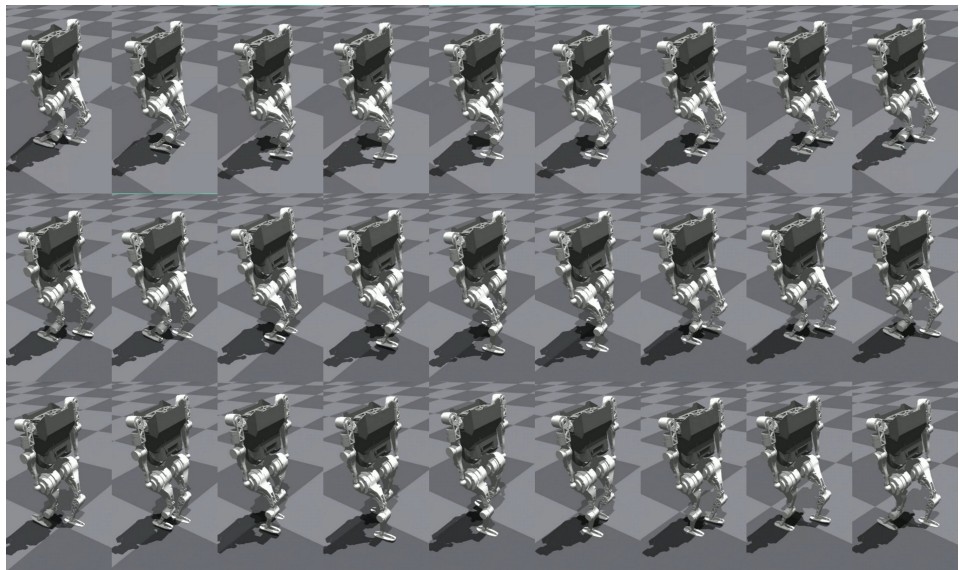

Figure 7: Diverse initial poses in simulation: A testament to our algorithm's robustness and stability.

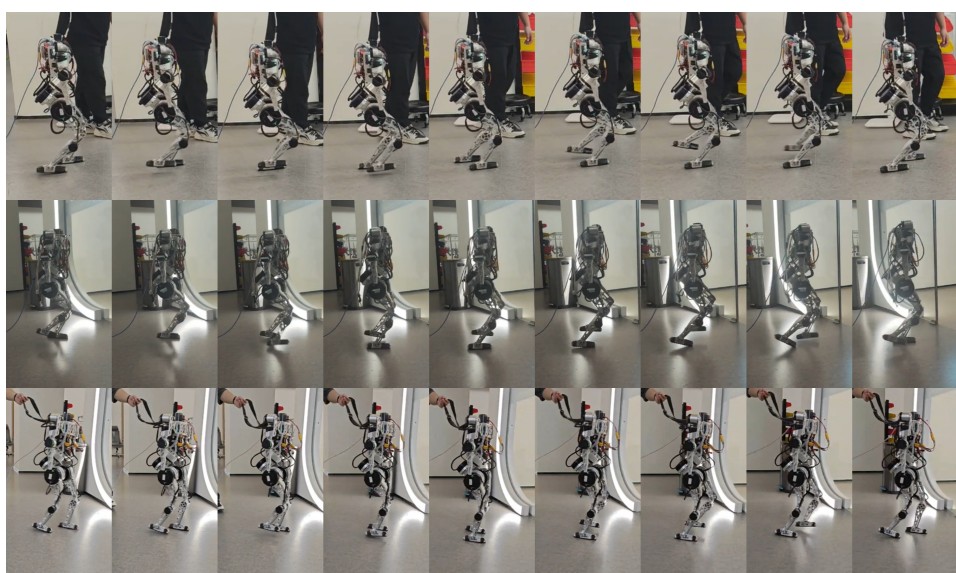

Figure 8: Simulated locomotion across diverse initial poses: Demonstrating algorithmic adaptability in sim-to-real transitions.

# D  REAL-MACHINE EXPERIMENTATION

We conducted multiple experiments on the actual machine, testing our policy on various terrains, including grasslands, wire-strewn ground, slopes, and stairs. Our robot's traversal across grasslands is a testament to its ability to handle the soft and unpredictable ground, while its passage through areas with wireless showcases its resilience against obstacles that could impede movement. The robot's ascent on slopes highlights its dynamic balance and the algorithm's capacity to adjust to inclines that require precise foot placement and torque control. Most notably, the robot's ability to climb and descend stairs indicates our algorithm's advanced control mechanisms, ensuring that each step is calculated for maximum efficiency and safety. The images reveal a robot that is not just mobile but one that can adapt to and stabilize on a wide array of environmental conditions, thereby proving the algorithm's robustness and stability in a comprehensive sim-to-real context.

These visual records are more than just demonstrations of our robot's physical capabilities; they are evidence of the sophisticated algorithms that enable it to interact intelligently with its surroundings, providing a solid foundation for further research and development in robotics.

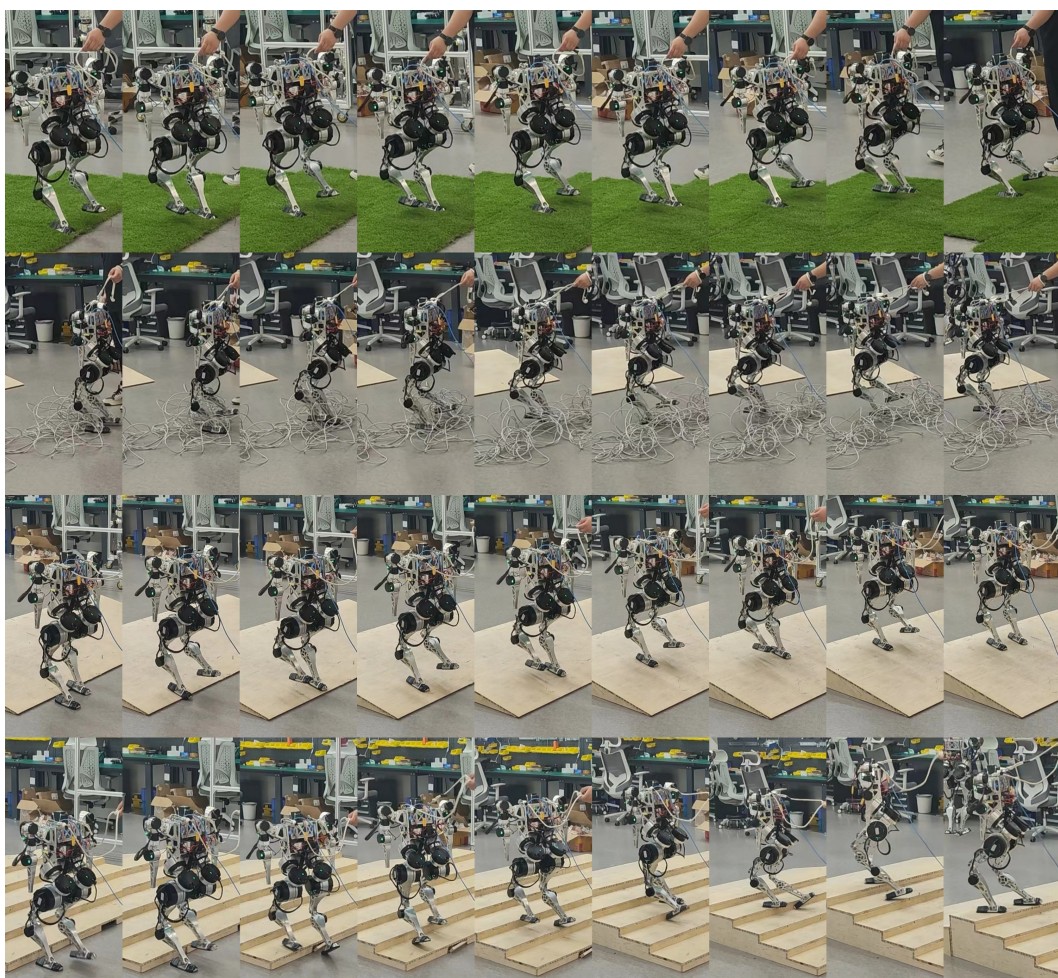

Figure 9: The sequence of images presented illustrates the diverse terrains our robot navigates with proficiency, ranging from the soft contours of grasslands to the challenging unevenness of wire-strewn areas, the inclines of varying gradients, and the ascents and descents of stairs. Each scenario, captured in a vertical progression from top to bottom, demonstrates not only the robot's adaptability but also its ability to maintain equilibrium on surfaces that demand different levels of traction and stability.

