# OpenReview forum: "Adapting Humanoid Locomotion over Challenging Terrain via Two-Phase Training"
_robot-learning.org/CoRL/2024/Conference — CoRL 2024_

### Official Review · Reviewer_RdfP · 2024-07-20

**Originality:** 3
**Technical Quality:** 4
**Clarity Of Presentation:** 4
**Potential Impact:** 3
**Recommendation:** 4
**Confidence:** 3

**Review:**

The paper presents a curriculum-like approach for training humanoid policies. The paper presents impressive results that are deployed on the physical robot. In addition, the organization of the work was well done and made it easy to follow the paper. The background literature motivates the work well and provides a nice overview of previous approaches and where this work fits.

In terms of weaknesses, there are a few that the paper should address. First, how does the paper change with respect to the PD controller and variability on the controller when transferred to the physical robot? As practitioners it would be good to see how much low-level control depends on the success of the approach. Some intuition for what the specific change of curriculum training should be in simulation. For instance, how quick of a change in difficulty would the approach fail or the learned policy become unstable or lose performance? This would be interesting as a means to provide intuition on how quickly the policy can adapt as difficulty is increased.

**Quality Of The Limitations Section:**

1

**Questions For Rebuttal:**

Failure modes for the approach are the main questions that would improve the contribution of this work. In particular, where does the approach fail and why is something that is missing from this work.

**Robotics Focus:**

4

**Summary Of Paper:**

The paper presents a method for humanoid robots to adapt over different terrains through the use of two-phased training.

**Summary Of Recommendation:**

The work is missing a clear overview of limitations which would improve the quality of the paper.

---

### Official Review · Reviewer_rN9h · 2024-07-21
**Strong results but missing ablations**

**Originality:** 3
**Technical Quality:** 3
**Clarity Of Presentation:** 4
**Potential Impact:** 3
**Recommendation:** 3
**Confidence:** 4

**Review:**

- The proposed method extends existing method in an important way to achieve superior performance compared to baseline methods.
- The paper will likely have a reasonable impact given the increased importance of humanoid research and that locomotion and stability is central to any downstream research on humanoids.
- The direction taken in the paper of learning in simulation via RL and then zeroshot transfer to real for locomotion in itself is not particularly new, however, there is no compelling recipe widely used for humanoids, hence making the contribution of this paper important.
- The push recovery in the video results is the most impressive and I would suggest the authors show more video and simulation based quantitative results of push recovery which is known to be a harder task for bipeds than quadrupeds.
- The paper is well written, easy to follow and understand. The figures are done well and its clear to understand the point of the method figure and the result tables.
- Some ablations that validate the importance of the algorithmic design are either missing from the comparison tables, or not clearly explained. More details below on this.
- The paper is lacking a limitations section and only superficially mentions "no vision" as the only limitation. A more detailed discussion on the limits of the current approach (like the amount of simulation tuning required, the need to pick parameters of domain randomization which cannot be too aggressive, stability and reproducibility of the proposed method), would help a lot.

**Quality Of The Limitations Section:**

1

**Questions For Rebuttal:**

- Estimator loss has two components. Could we ablate the importance of LVAE loss?
- Figure 4: Could the authors define FHM used in the legend (point to where in the paper its defined)?
- Are the terrains included in Phase one of the training or only in phase 2? If only included in phase 1, authors should include a comparison with terrains included in Phase 1.
- L171: The sentence is not clear, based on what finding?
- L237-L239: “This refined passage employs a more formal and academic 238 tone, with a clear and structured explanation of the experimental setup, observation, and the comparative efficacy of the proposed method against existing ones.” Has the passage been written by a language model?
- Please address the limitations more thoroughly with discussion on things like: amount of simulation tuning required, the need to pick parameters of domain randomization which cannot be too aggressive, stability and reproducibility of the proposed method, etc.

**Robotics Focus:**

4

**Summary Of Paper:**

The paper proposes a two phase RL training in simulation with zero shot transfer to real for humanoid locomotion. The method relies on a learned estimator and a curriculum in terrain (two phase training) as well as velocity sampling to improve sim based RL learning. The proposed method shows high robustness in the real world and quantitatively superior performance to baselines.

**Summary Of Recommendation:**

The proposed method shows very compelling results in the real world on a humanoid robot. The experiments could be improved to make them more thorough

---

### Official Review · Reviewer_61BH · 2024-07-22

**Originality:** 2
**Technical Quality:** 2
**Clarity Of Presentation:** 4
**Potential Impact:** 2
**Recommendation:** 3
**Confidence:** 4

**Review:**

This paper has nice real-world results, although I think prior work exists which shows superior results albeit on different hardware. The method itself seems to be a minor change to DreamWaq (1) adding velocity prediction to the estimator and (2) using observation history instead of just a single observation. This implies that the method has limited novelty. Moreover, the authors have not compared against some important baselines, and it is unclear to me what the benefit of this method is over the ones proposed in prior work.

- Why have the authors not compared against A-RMA [1], which also uses privileged observations in sim, and has shown results on real biped robots?
- How many frame stacks does the baseline get? Did the authors also try RNNs which have a much longer context length and have shown success in prior work [2]?

I would encourage the authors to position their paper better in relation to prior work and add more thorough evaluation.

[1] Kumar, Ashish, et al. "Adapting rapid motor adaptation for bipedal robots." 2022 IEEE/RSJ International Conference on Intelligent Robots and Systems (IROS). IEEE, 2022.
[2] Siekmann, Jonah, et al. "Blind bipedal stair traversal via sim-to-real reinforcement learning." arXiv preprint arXiv:2105.08328 (2021).

**Quality Of The Limitations Section:**

1

**Questions For Rebuttal:**

- In the video results, when going up / down the steps it is hard to tell whether the operator is applying a stabilizing force to the robot through the cord. Is the robot able to do the stairs reliably without support? Can the authors generate a video where the robot is held by a loose cord so that no force is being applied on it?
- Is the only difference from DreamWaQ that the estimator gets a history of observations instead of a single observation?

**Robotics Focus:**

4

**Summary Of Paper:**

This paper extends studies the problem of robust blind biped locomotion. They extend prior work (DreamWaq) - an estimator network is trained to estimate velocities and map observation history to latent latent states. These condition a downstream policy which is trained via RL in simulation with domain randomizaton. They demonstrate locomotion on a real robot on small stairs, slopes, grass and flat ground with some perturbances.

**Summary Of Recommendation:**

The paper has nice real world results, however, I vote to weak reject this paper because of limited novelty in both results and method + insufficient comparison to prior work

---

### Author Rebuttal · Authors · 2024-08-08

Dear Area Chair and esteemed Reviewers, we extend our heartfelt thanks for the considerable time and effort you have invested in providing our work with intriguing and valuable suggestions.

In response to the reviewer 61BH's feedback:
* We have added a comparison with **A-RMA** in Section 4, specifically in the Multi-task, Stability, and Commands Tracking sections.
* Additionally, we have clarified the number of frame stacks used in our experiments in lines 43, 53, and 160. We have also included a comparison with the RNNs network in the official comment section below.
* Following the reviewer's suggestion, we have provided the latest video of ascending and descending stairs, demonstrating our capability to perform these tasks even with a loose cord.
* Moreover, in the official comments, we reiterated the **novelty of our work**, which lies not only in the estimator with history stacks but also in the design of our **two-phase training approach**.

Regarding the reviewer rN9h's recommendations:
* we have added some **ablation experiments**. First,we have included a comparison with the version without LVAE to highlight its indispensability.
* Based on the reviewer's advice, we have removed the reference to FHM in Figure 4 and explained this change.
* We have also added a comparison with Phase 1 in the command velocity tracking section to substantiate the necessity of Phase 2.
* Naturally, we have made revisions to line 171 and eliminated the sentences at lines 237-239 based on the reviewer's feedback on the paper's logic.
* Finally, heeding the reviewer's suggestion, we have added a **limitations section** detailing the limitations related to target robot reward shaping and domain randomization, with relevant examples.

In light of the reviewer RdfP's suggestions:
* We have clarified the frequencies of our PD controller and policy and indicated that future research will be conducted on this matter.
* Furthermore, in response to the reviewer's inquiry about failure modes, we have provided answers and expanded our **limitations section** accordingly.

---

### Decision · Program_Chairs · 2024-09-04

**Decision:**

Accept

**Comment:**

This paper introduced a two-phase training approach to blind biped locomotion. They used an estimator network to predict velocities and map observation history to latent states. They then used these latent states to condition an RL policy trained in large-scale simulation environments and transfer the policy to the physical robot in real-world terrains. This paper received mixed initial ratings, with one weak reject and two weak accepts. The reviewers pointed out the strengths of this paper, particularly 1) the use of curriculum learning is an interesting idea, and 2) the real-robot deployments showed the robustness and practicality of this approach. In the meantime, they also mentioned several weaknesses, including 1) limited novelty: this work seems to be an incremental improvement over previous work, particularly DreamWaQ, and 2) experimental validation: comparisons to relevant methods, such as A-RMA, are missing and the experiments lack thorough ablation studies to validate the importance of specific components of the algorithm. In addition, all three reviewers found that the limitations were not well addressed. Overall, the AC thinks this work has good promises, but the reviewers' criticisms are valid and must be addressed before this paper passes the bar of publication.

**Post-rebuttal update**

The authors successfully addressed the concerns raised by the three reviewers. Toward the end of the rebuttal period, the reviewers had three unanimously positive recommendations. The AC felt pleased to see the positive changes, which improved the quality of this paper, and recommended accepting it for CoRL.